# The Role and Pathogenesis of Tau Protein in Alzheimer’s Disease

**DOI:** 10.3390/biom15060824

**Published:** 2025-06-05

**Authors:** Xiaoyue Hong, Linshu Huang, Fang Lei, Tian Li, Yi Luo, Mengliu Zeng, Zhuo Wang

**Affiliations:** 1Department of Laboratory Medicine, Zhongnan Hospital of Wuhan University, Wuhan 430071, China; hongxy1873@foxmail.com (X.H.); huanglinshu2001@163.com (L.H.); luoyi929@aliyun.com (Y.L.); 2Hubei Provincial Clinical Research Center for Molecular Diagnostics, Zhongnan Hospital of Wuhan University, Wuhan 430071, China; 3Medical Science Research Center, Zhongnan Hospital of Wuhan University, Wuhan 430071, China; leifang@whu.edu.cn; 4Department of Neurology, Zhongnan Hospital of Wuhan University, Wuhan 430071, China; litian@whu.edu.cn

**Keywords:** Alzheimer’s disease, tau, post-translational modifications, hyperphosphorylation, acetylation, synaptic dysfunction, cognitive impairment

## Abstract

Alzheimer’s disease (AD), a predominant neurodegenerative disorder, is clinically characterized by progressive cognitive deterioration and behavioral deficits. An in-depth understanding of the pathogenesis and neuropathology of AD is essential for the development of effective treatments and early diagnosis techniques. The neuropathological signature of AD involves two hallmark lesions: intraneuronal neurofibrillary tangles composed of hyperphosphorylated tau aggregates and extracellular senile plaques containing amyloid-β (Aβ) peptide depositions. Although Aβ-centric research has dominated AD investigations over the past three decades, pharmacological interventions targeting Aβ pathology have failed to demonstrate clinical efficacy. Tau, a microtubule-associated protein predominantly localized to neuronal axons, orchestrates microtubule stabilization and axonal transport through dynamic tubulin interactions under physiological conditions. In AD pathogenesis, however, tau undergoes pathogenic post-translational modifications (PTMs), encompassing hyperphosphorylation, lysine acetylation, methylation, ubiquitination, and glycosylation. These PTM-driven alterations induce microtubule network disintegration, mitochondrial dysfunction, synaptic impairment, and neuroinflammatory cascades, ultimately culminating in irreversible neurodegeneration and progressive cognitive decline. This review synthesizes contemporary advances in tau PTM research and delineates their mechanistic contributions to AD pathogenesis, thereby establishing a framework for biomarker discovery, targeted therapeutic development, and precision medicine approaches in tauopathies. This review synthesizes contemporary advances in tau PTM research and delineates their mechanistic contributions to AD pathogenesis, thereby establishing a solid theoretical and experimental basis for the early diagnosis of neurodegenerative diseases, the discovery of therapeutic targets, and the development of novel therapeutic strategies.

## 1. Introduction

Alzheimer’s disease (AD) is a widespread neurodegenerative disorder that represents a substantial threat to the health and quality of life of elderly individuals worldwide. According to the “World Alzheimer’s Report 2018” published by Alzheimer’s Disease International (ADI), the global prevalence of dementia is estimated to affect at least 50 million individuals. This figure is projected to rise to 152 million by 2050, with approximately 60% to 70% of these cases attributable to AD. Tau protein, a microtubule-associated protein predominantly localized to neuronal axons, serves critical functions in maintaining microtubule stability, promoting microtubule assembly and regulating axon transport [1,2,3]. In AD pathogenesis, tau proteins undergo abnormal post-translational modifications (PTMs), including but not limited to hyperphosphorylation, acetylation, ubiquitination, glycosylation, and methylation [4]. These PTMs interact to form a complex regulatory network that jointly acts on the structure and function of tau protein, as well as its interactions with other molecules, profoundly influencing processes such as the formation of neurofibrillary tangles, signal transduction between neurons, and neuroinflammatory responses, ultimately promoting the deterioration of the disease. This review employs a narrative approach to synthesize our team’s accumulated knowledge regarding the relationship between tau proteins and AD. The aim of this review is to analyze and summarize the types, sites, and enzymes of tau PTMs and the mechanisms of abnormal tau protein accumulation that trigger AD, including mitochondrial dysfunction, synaptic plasticity impairments, and glia-mediated neuroinflammation. In pathological conditions, understanding the mechanisms and significance of tau toxicity is crucial for elucidating its role in exacerbating neurodegeneration and cognitive impairment in AD. This insight provides a solid theoretical and experimental foundation for early diagnosis, the discovery of potential therapeutic targets, and the formulation of novel treatment strategies.

## 2. Search Strategy

This review focuses on the role and pathogenesis of tau proteins in AD. Articles for this review were searched via PubMed up to April 2025. The following keywords were used: AD, tau, phosphorylation, acetylation, ubiquitination, truncation, neuroinflammation, mitochondrial damage, synaptic dysfunction, and cognitive impairments. The studies were selected and cited prioritizing their relevance to the topic of this review. Subsequently, their authority, high impact factor, and timeliness were comprehensively evaluated.

## 3. Basic Structure of Tau Protein

Tau protein is encoded by the microtubule-associated protein tau (MAPT) gene, mapped to human chromosome 17 [5,6]. This gene encompasses over 100 kilobases and contains 16 exons. Alternative splicing of these exons generates six principal tau protein isoforms in the human central nervous system (CNS). Structurally, tau is categorized into several major domains. The amino-terminal projection domain (N-terminal domain) corresponds to the proline-rich projection domain, which constitutes approximately two-thirds of the protein’s total length. This domain includes an alkaline proline-rich motif (P-region) and acidic amino-terminal inserts. The projection domain interacts with other cytoskeletal proteins, extending outward from the microtubule surface to contact other cytoskeletal components and the cell membrane, thereby playing a critical role in maintaining axonal stability [7]. The carboxy-terminal microtubule-binding domain (C-terminal domain) represents the microtubule-binding domain, which is the primary functional domain of the tau protein. This domain is responsible for binding to tubulin to stabilize the microtubule structure. It consists of a carboxyl-terminal region composed of repetitive peptide sequences and amino acid residues that bind to tubulin. Specifically, there are four main repeat fragments, designated as R1, R2, R3, and R4 (Figure 1). These repeats specifically interact with tubulin, promoting microtubule assembly and stability, and are involved in the regulation of axonal transport and synaptic plasticity in neurons. Tau proteins are expressed primarily in the CNS and comprise six isoforms containing either three (3R-tau) or four (4R-tau) repeat domains that mediate microtubule binding, thereby regulating microtubule stability [8,9]. A study demonstrated that the three-dimensional structure of the tau protein, as predicted by the ab initio model, is capable of undergoing various PTMs, thereby providing novel insights into the understanding of tau protein structure and its associated pathogenesis [10].

## 4. PTMs of Tau Protein

In the brains of AD patients, pathological PTMs including hyperphosphorylation, glycosylation, acetylation, methylation, and ubiquitination accumulate on tau (Table 1, Figure 2). At present, techniques such as Western blotting, immunohistochemistry [11], immunofluorescence [12], proteomics, immunomagnetic exosomal PCR [13], affinity purification-mass spectrometry [14], and liquid chromatography-mass spectrometry [15] have enabled an accurate detection of the PTMs of tau proteins. It is of great significance to further study the PTMs of tau proteins. It helps us to reveal the pathogenesis of tau protein disease at the molecular level, and clarify how abnormal modifications lead to neuropathy through a step-by-step process, providing critical clues for a comprehensive understanding of the pathological process of neurodegenerative diseases. Targeting the rate-limiting enzymes or modification sites in the abnormal modification process of tau proteins is expected to develop therapies that can effectively stop disease progression or even reverse the disease, bringing new hope to patients with tau protein disease. Next, we will focus on the PTMs of tau proteins.

### 4.1. Tau Phosphorylation

Phosphorylation is recognized as an important PTM of tau proteins because it is involved in physiological and pathological states [16,17]. In healthy brains, tau regulates microtubule stability through a reversible process of phosphorylation and dephosphorylation [2,18]. Phosphorylation is required for tau proteins to bind to microtubules. However, the hyperphosphorylation of tau proteins leads to their dissociation from microtubules and subsequently results in aggregation. In the brains of AD patients, the phosphorylation process of tau proteins becomes profoundly dysregulated, representing a critical event in AD pathogenesis. The phosphorylated state of tau proteins, in turn, is influenced by kinase activity levels and the balance between kinases and phosphatases within neurons. PTM mapping of tau proteins obtained from the brains of AD patients reveal phosphorylation sites that are absent under normal conditions. Some major pathological sites include AT8 (pS202/pT205), AT100 (pT212/pS214), AT180 (pT231/pS235), PHF1 (pS396/pS404), pS356, pY394, pT403, pS409, and pS422 [19,20]. Most of these sites are located in the repetitive and lateral regions of tau. Modifications at specific sites may induce tau aggregation by interfering with charge distribution and altering intramolecular interactions.

The phosphorylation of tau proteins is precisely regulated by various protein kinases and phosphatases, maintaining a dynamic equilibrium [21]. Protein kinases are classified into three groups: proline-directed protein kinases (PDPKs), non-proline-directed protein kinases (non-PDPKs), and protein tyrosine kinases (PTKs). In the brain of AD patients, the expression of multiple kinases is increased. Glycogen synthetase kinase-3β (GSK-3β) [22,23,24], and cyclin-dependent kinase 5 (CDK5) [25,26] are two kinases that have been extensively studied. They co-localize with pathological tau and can phosphorylate tau at multiple sites. For instance, GSK3β and CDK5 can phosphorylate Ser202, Thr212, Ser214, and other sites. In mouse models, the inhibition of CDK5, the downregulation of GSK3β activity, or the use of GSK3β inhibitors can effectively reduce tau phosphorylation levels. In addition to GSK3β and CDK5, p38 mitogen-activated protein kinase (p38 MAPK) [27]; the c-Jun kinase family (JNK) and other PDPKs; Ca^2+^/non-PDPKs such as calmodulin-dependent protein kinase II (CaMKII) and cyclic adenosine phosphate dependent protein kinase A (PKA) [28]; and PTKs such as proline-rich tyrosine kinase 2 (Pyk2), and lymphocyte-specific protein tyrosine kinase (Fyn). They are also implicated in the regulation of tau protein phosphorylation.

Phosphatase plays a crucial role in the dephosphorylation process of tau proteins, with key enzymes including PP1, PP2A [29,30], PP2B, PP2C, and PP5, which mediate tau dephosphorylation. In the brains of AD patients, PP2A activity is reduced by 20% in gray matter and 40% in white matter, and PP2A inactivation involves multiple mechanisms, such as PTMs of the PP2A catalytic structure, and decreased mRNA and protein expression of the catalytic and regulatory subunits. Additionally, there are elevated levels of endogenous PP2A inhibitors, such as I1PP2A, I2PP2A [31], and CIP2A [32]. Studies have demonstrated that inhibiting PP2A leads to tau hyperphosphorylation and memory deficits, while silencing I2PP2A or inhibiting CIP2A can reduce tau hyperphosphorylation and improve memory deficits [33].

The phosphorylation sites of tau proteins may play opposing roles in physiological and pathological processes. For instance, phosphorylation at the Ser202 and Thr231 sites of tau promotes its aggregation, neurofibrillary tangle formation, and synaptic dysfunction [34]. Additionally, a study demonstrated that a humanized monoclonal antibody targeting the phosphorylation of Thr231 in tau could reduce neurofibrillary tangle formation and tau seeding activity while enhancing synaptic and cognitive functions [35]. These findings suggest that distinct tau phosphorylation sites play differential roles in tau-related pathology and cognitive impairment. Consequently, selecting the appropriate antigenic epitope when developing therapies targeting tau phosphorylation may enhance both the safety and efficacy of such interventions.

### 4.2. Tau Acetylation

The acetylation of tau proteins is a significant PTM that plays a pivotal role in the pathogenesis of AD. It has been found that tau proteins can undergo acetylation at multiple lysine (Lys) sites, with particular attention given to the acetylation of K274 and K281 sites in AD research [36]. In the early stages of AD, the acetylation of tau proteins at K274 and K281 is significantly increased, and is even more pronounced in patients with advanced AD and severe dementia.

The acetylation of tau proteins has many effects on its function and structure. From a functional perspective, acetylation disrupts the normal interaction between tau proteins and microtubules, leading to the dissociation of tau from microtubules, thus destroying the stability of microtubules, impairing intracellular transport, and affecting cytoskeletal maintenance. Structurally, acetylation alters the molecular conformation of tau proteins, making them more susceptible to aggregation. A collaborative study demonstrated that the acetylation of simulated tau proteins at the K274 and K281 sites led to a decrease in mitochondrial biogenesis, a decrease in the expression of mitochondrial fusion proteins, and an increase in mitochondrial dysfunction [37]. Consequently, the cognitive impairment of mice was exacerbated. This suggests that tau protein acetylation further exacerbates the pathological progression of AD by affecting mitochondrial function. In addition, the acetylation of tau proteins may also interact with other PTMs, such as phosphorylation, to collectively regulate the function and fate of tau, with the specific molecular mechanisms warranting further investigation [38].

The acetylation modification of tau proteins mainly occurs on lysine residues, and this modification is a dynamic and reversible process, co-regulated by acetyltransferase and deacetylase. The p300/CBP-associated factor (PCAF) and p300/CREB-binding protein (p300/CBP) are the main acetyltransferases responsible for catalyzing the acetylation of tau proteins [39,40,41]. The PCAF specifically recognizes a lysine residue on tau proteins and transfers the acetyl group from acetyl-CoA to the ε-amino group of this residue, thereby mediating the acetylation of tau proteins. In vitro, the co-incubation of PCAFs and tau proteins results in the detection of the acetylation of tau proteins at multiple lysine sites, and the degree of acetylation increases with the increase in PCAF concentration. p300/CBP also exhibits a similar catalytic effect, which can acetylate tau proteins in different cellular environments and regulate tau protein function. In addition, mammalian tau proteins possess intrinsic enzymatic activity capable of catalyzing self-acetylation. Tau utilizes catalytic cysteine residues in the microtubule-binding domain to facilitate the acetylation of lysine residues, thus suggesting a mechanism similar to that employed by MYST-family acetyltransferases [42]. The repeat domain K18 mediates the acetyltransferase function of tau proteins [43]. In addition to acetylating itself, tau can also acetylate STAT1, and other proteins associated with the AD pathway [44].

Histone deacetylases (HDACs) catalyze the reverse reaction, removing acetyl groups and deacetylating tau proteins. In addition, sirtuin 1 (SIRT1) and sirtuin 2 (SIRT2) are key deacetylases involved in tau protein deacetylation. Researchers have reported the deacetylating activity of SIRT1 on the acetylated Lys174 (K174) of tau in tauP301S transgenic mice with a brain-specific SIRT1 deletion. SIRT1 deficiency leads to exacerbation of premature mortality, synapse loss, and behavioral disinhibition in tauP301S transgenic mice [45,46]. SIRT2 also has the ability to deacetylate tau proteins, but its localization and activity regulation in a cell differ from that of SIRT1 [47,48]. Both enzymes play critical roles in maintaining the homeostasis of tau protein acetylation levels. In the pathological state of neurodegenerative diseases, the activity of acetyltransferase and deacetylase is unbalanced, resulting in abnormal acetylation modifications of tau proteins, which in turn impairs the function of tau proteins.

Tau^KQ^high mice expressing human tau with lysine-to-glutamine mutations that mimic acetylation at K274 and K281 have been previously shown to exhibit memory impairments and deficits in long-term potentiation [36]. The interesting observation that the hypothalamus of tau^KQ^high mice exhibits significantly increased axonal neurodegeneration contrasts the previously observed neuroprotective effects in this region following traumatic brain injury [49]. The monoclonal antibody Y01 specifically targeting tau protein Lys280 has demonstrated significant therapeutic effects on tau protein accumulation, reproduction, and cognitive deficits in tau transgenic mouse models [50]. Therefore, targeting tau acetylation represents a promising, novel therapeutic strategy for treating tauopathies in humans.

### 4.3. Tau Methylation

The methylation modification sites of tau proteins are primarily concentrated on the Lysine (Lys) residues within its microtubule-binding domain (MBD) [51]. Methylation modifications of tau proteins significantly influence their binding ability with microtubules. Studies have shown that methylation at certain sites of tau proteins, such as K280 methylation, can reduce the affinity between tau proteins and microtubules, leading to the disassociation of tau proteins from the microtubules. Methylation at certain sites, such as K321, can promote tau aggregation. K254 methylation may exert neuroprotective effects by maintaining the normal conformation of tau proteins and preventing the abnormal aggregation of tau. Methylation at certain sites, such as K353, can enhance the transport of tau proteins from axons to cell bodies and dendrites, resulting in the accumulation of tau proteins in cell bodies and dendrites [52]. This abnormal intracellular localization may impair the normal function of tau proteins, disrupting signal transduction within neurons and further aggravating the damage of neurons. In the study of AD, through the analysis of brain tissue samples of patients, it was found that the methylation level of tau proteins at certain sites was significantly lower than that of the normal controls, and this reduction in methylation was closely associated with abnormal tau aggregation and the formation of neurofibrillary tangles [53].

Under the action of lysine methyltransferase or arginine methyltransferase, tau methylation occurs on several LYs and some arginine residues [52,54]. However, the specific methyltransferases involved in tau modification remain poorly understood. The role of methyltransferase SETD7 in the methylation of K130 and K132 and its importance in the nuclear localization of tau proteins have been reported [55]. Most methylation sites are located within the tau microtubule-binding region. Tau methylation does not affect tubulin polymerization, and its aggregation tendency decreases with increasing methylation. Studies have shown that the number of monomethylation sites increases with age and the progression of AD. In healthy brains, soluble tau proteins also contain methylated arginine sites. Recent studies suggest that tau methylation plays a role in normal tau function and the formation of pathologically paired helical filaments (PHFs) in AD patients. The arginine residues R126, R155, and R349 are known to be methylated in both normal and pathological tau [56]. Tau methylation can compete with other PTMs such as acetylation and ubiquitination. Tau methylation levels at K254 in PHFs exceed their ubiquitination levels, hindering the clearance of tau aggregates by the ubiquitin proteasome system (UPS) in AD patients. Another lysine residue, K290, was found to be ubiquitinated in aggregated tau, whereas it was methylated under normal conditions [56,57,58]. The state of PTMs at these sites determines the fate of tau proteins in terms of their function and stability [53]. Tau phosphorylation at S262 was found to occur more frequently along with methylation at K267 [57]. A PHF-derived tau contains fewer methylation sites than normal tau. Two methylation sites for K24 and K44 are located near the caspase and calpain cleavage sites, while other methylation sites produce fragments that are prone to aggregation [4,59,60]. Although research on the direct effects of methylation on tau function and aggregation remains limited, the existing evidence indicates it may play a crucial role in determining the fate of tau.

### 4.4. Tau Ubiquitination

Studies have shown that the tau protein possesses multiple ubiquitination sites, and its ubiquitination modification is involved in the degradation and metabolism of tau proteins [61]. Under physiological conditions, the ubiquitination modification of tau proteins serves as a signal for the recognition and degradation of abnormal or excess tau by the proteasome, thus maintaining the stability and normal function of the tau protein level in a cell. However, in the brains of AD patients, the ubiquitination process of tau proteins is abnormal. On the one hand, an abnormally phosphorylated tau protein may interfere with its normal ubiquitination modification, impairing its effective degradation and leading to the gradual accumulation of tau proteins in the cell. On the other hand, changes in the activity or expression levels of the enzymes involved in the tau ubiquitination process (such as ubiquitin ligase, etc.) will also influence the efficiency of tau ubiquitination.

One strategy to enhance tau clearance is to target protein degradation, thereby promoting tau degradation in the UPS and the autophagy lysosome pathway (ALP) [62]. The UPS is a highly conserved intracellular mechanism that maintains protein homeostasis and eliminates damaged, misfolded, and mutated proteins in the cytoplasm and nucleus [63,64,65]. UPS involves the covalent attachment of multiple ubiquitin molecules to the protein substrate and the degradation of the ubiquitinated protein by the proteasome complex [66]. Ubiquitin activator E1 first forms a high-energy thioester bond with the C-terminal of ubiquitin molecules under the action of ATP, thereby activating ubiquitin molecules. The activated ubiquitin molecule is transferred to the ubiquitin binding enzyme E2, where it forms a thioester bond with the ubiquitin molecule through the cysteine residue at its active site. Ubiquitin ligase E3 plays a key role in specific substrate recognition during ubiquitination; it is able to recognize tau proteins and transfer the ubiquitin molecules bound to E2 to the lysine residues of a tau protein to form the ubiquitin–tau protein complex. In this process, E3 interacts with the tau protein through its specific domain, ensuring that the ubiquitin molecule is precisely attached to a specific site in the tau protein. Multiple ubiquitin molecules are sequentially linked to form polyubiquitin chains, which ultimately tag tau proteins for recognition and degradation by the proteasome. The ubiquitinated proteins can be recognized by the intrinsic ubiquitin receptors of the proteasome or shuttle factors, such as p62/SQSTM1, which contain both a ubiquitin-associated domain and a domain that binds to the proteasome [67].

A variety of E3 ligases are involved in the ubiquitination of tau, such as the tumor necrosis factor receptor-associated factor 6 (TRAF6) and neural precursor cell expressed, developmentally downregulated 4 (NEDD4) [68,69,70]. TRAF6 specifically recognizes tau proteins and promotes their ubiquitination modification. In cell-based experiments, the overexpression of TRAF6 resulted in a significant increase in tau ubiquitination, whereas TRAF6 knockdown decreased tau ubiquitination. NEDD4 also participates in the ubiquitination process of tau proteins by interacting with a specific domain of tau proteins to mediate the conjugation of ubiquitin molecules and regulate the ubiquitination level of tau proteins. These E3 ligases play a fine regulatory role in the ubiquitination of tau under various cellular environments and both physiological and pathological conditions [71,72].

Studies have shown that a high expression of the X-chromosome associated ubiquitin-specific peptidase 11 (USP11) promotes tau deubiquitination and stabilizes tau, thereby preventing its normal clearance and accelerating its pathological aggregation, which leads to cognitive dysfunction in AD mice [73,74]. However, the deletion of the USP11 gene can significantly improve tau pathology and cognitive function impairment in AD mice [73,74]. This suggests that tau deubiquitination abnormalities play a crucial role in the pathogenesis of AD, and the inhibition of USP11-mediated tau deubiquitination to improve AD pathology is expected to provide a new strategy for AD treatment. In addition, the abnormal ubiquitination of tau proteins may also be related to the formation and stability of neurofibrillary tangles, but the specific molecular mechanism remains to be further explored [75]. The ubiquitination of tau proteins is a complex cascade reaction process involving the participation of several key proteins.

### 4.5. Tau Glycosylation

Glycosylation modification has an important effect on the function of tau proteins, influencing their solubility, stability, and propensity to aggregate. Abnormal glycosylated tau protein molecules will form extensive cross-linking, resulting in reduced solubility and an increased tendency to aggregate into neurofibrillary tangles. Glycosylation modification may also disrupt the normal function of neurons by affecting the interaction of tau proteins with other molecules. N-glycosylation has been shown to affect the stability, solubility, folding, structure, and function of various proteins, as well as oligomerization and fibril formation [76,77]. Extensive research has successfully identified multiple glycosylation sites and various tau protein modification types. Within the amino acid sequence of the tau protein, several serine (Ser), threonine (Thr), and asparagine (Asn) residues have been identified as glycosylation sites. Studies have demonstrated that specific sites, such as Ser-409, Ser-412, and Ser-413, are subject to O-GlcNAc glycosylation modifications. This type of modification plays a crucial role in regulating tau protein function by inhibiting phosphorylation at certain sites, thereby influencing its microtubule-binding capacity and aggregation behavior within cells. For instance, when O-GlcNAc glycosylation occurs at the Ser-409 site, the phosphorylation level at this site decreases, which alters the interaction between the tau protein and microtubules, leading to reduced microtubule stability. Furthermore, sites such as Asn-291 and Asn-368 of the tau protein have been found to undergo N-glycosylation modifications. N-glycosylation typically involves more complex glycan structures, and the presence of these glycans may alter the spatial conformation and biological activity of the tau protein [77,78,79].

### 4.6. Tau Truncation

Tau protein truncation refers to the process by which tau proteins are cleaved by specific proteases, resulting in tau protein fragments of varying lengths. Tau truncation occurs in the early stages during the development of human AD and other tauopathy dementias. Notably, tau cleavage, especially within its N-terminal projection domain, has the potential to independently drive neurodegeneration [80].

It has been observed that a variety of proteases participated in the truncation process of tau proteins, such as calpain and caspase. These proteases exhibit abnormal activity in the brains of AD patients, and they can specifically cleave specific peptide bonds of tau proteins, resulting in truncated fragments of varying lengths. For example, caspase-3 can cleave tau proteins at Asp421, producing a C-terminally truncated tau fragment [81,82,83]. These truncated tau protein fragments, due to their altered structure, have a stronger tendency to aggregate and are more likely to form neurofibrillary tangles than normal tau proteins, thus causing greater damage to neurons. In addition, truncated tau protein fragments may also interfere with the normal function of tau, disrupt microtubule stability, and affect neuronal axonal transport and signal transduction. Tau protein truncation may also trigger immune and inflammatory responses within cells, further aggravating neuronal damage and death. Consequently, tau protein truncation represents a critical step in the pathogenesis of AD, and the in-depth study of its molecular mechanism is of great significance for understanding the pathological process of AD and developing potential therapeutic strategies.

Asparagine endopeptidase (AEP), also known as delta-secretase, cleaves protein substrates at the C-terminal of asparagine and directly cleaves tau proteins at the N255 and N368 sites. AEP cutting tau at N368 eliminates the protein’s ability to bind to microtubules, thereby enhancing its aggregation ability and increasing its neurotoxicity. A Tau1-368 fragment has a stronger aggregation ability and neurotoxicity than full-length tau [84]. Among the truncated tau, Tau151-391 showed the highest pathological activity. The first 150 amino acids and the last 50 amino acids protect tau from pathological characteristics and their deletions facilitate pathological activities. Thus, the inhibition of tau truncation may represent a potential therapeutic approach to suppress tau pathology in AD and related tauopathies [85]. Our study indicated that the Tau (1-368) fragment is more robust than full-length tau in binding active STAT1, a BACE1 transcription factor, and that it promotes its nuclear translocation, which upregulates BACE1 expression and Aβ production. Notably, Aβ-activated SGK1 or JAK2 kinase phosphorylates STAT1 and induces its association with Tau (1-368) [86].

## 5. Mechanism of Abnormal Tau Protein Accumulation Triggering AD

Abnormal PTMs reduce the binding affinity between tau and microtubules, leading to an accumulation of free tau proteins in the cytoplasm. The accumulation of free tau proteins facilitates the formation of tau aggregates, which in turn induces mitochondrial dysfunction, impairs synaptic plasticity, and triggers glial-mediated neuroinflammation. In these pathological conditions, understanding the mechanisms and significance of tau toxicity is crucial for elucidating its role in exacerbating the neurodegeneration and cognitive impairment associated with AD.

### 5.1. Damage to the Neural Microtubule System

Under normal physiological conditions, a tau protein specifically binds to tubulin through its microtubule binding domain, promoting the assembly and stability of microtubules, and maintaining the normal morphology of neurons and axonal transport function [87]. However, in the brains of patients with AD, tau proteins undergo abnormal hyperphosphorylation with an increase in phosphorylation sites, resulting in a significant decrease in their affinity for microtubules. Studies have shown that multiple phosphorylation sites for tau proteins, such as Thr212, Ser214, Thr231, Ser235, and Ser262, will greatly reduce the ability of tau proteins to bind to microtubules after phosphorylation [88]. This decreased binding ability makes tau proteins unable to effectively maintain the stability of microtubules, resulting in microtubule disaggregation and a disruption of the neuronal cytoskeletal structure [89].

In addition to phosphorylation, the truncation of tau proteins also has a destructive effect on the neural microtubule system [90]. In the brain of AD patients, tau proteins are cleaved by caspase-6, calpain, and other proteases, generating truncated fragments of varying lengths [91]. Due to the loss of a specific domain, the binding affinity of these truncated fragments to microtubules is further reduced, and microtubule depolymerization is more likely to be caused. Additionally, the truncated fragment may interfere with the interaction between normal tau proteins and microtubules, disrupting the normal assembly and function of microtubules by competing with normal tau proteins for microtubule binding. It has been found that certain truncated fragments can aggregate, and these aggregates are deposited in neurons, further destroying the structure and function of neurons, and exacerbating axonal transport deficits [92,93].

### 5.2. Inducing Mitochondrial Damage

Tau proteins can affect mitochondrial function through multiple mechanisms. Tau proteins can interact with related proteins on the mitochondrial surface, interfering with normal mitochondrial transport [94]. Neurofibrillary tangles composed of hyperphosphorylated tau proteins can impair mitochondrial respiratory chain function. Research has demonstrated that in the brains of AD patients, the activities of mitochondrial respiratory chain complexes I, II, III, and IV are significantly reduced [95,96,97]. Tau protein aggregates have been shown to bind directly to complex I of the mitochondrial respiratory chain, thereby inhibiting its enzymatic activity. This inhibition disrupts electron transport and impairs proton gradient formation, ultimately leading to a reduction in ATP synthesis. Furthermore, mitochondria serve as one of the primary sources of intracellular reactive oxygen species (ROS) [98,99]. An abnormal aggregation and phosphorylation of tau proteins have been observed, coinciding with increased JNK activity. Additionally, mitochondrial morphology undergoes significant alterations, becoming shorter and fragmented, with the disruption of the mitochondrial cristae structure [100]. Upon JNK activation, the expression levels of proteins associated with mitochondrial fusion and fission were modulated; specifically, the expression of Mfn1, Mfn2, and OPA1 decreased, whereas the expression of Drp1 and Fis1 increased, thereby disrupting the mitochondrial dynamics equilibrium. Simultaneously, JNK activation upregulated the pro-apoptotic protein Bax and downregulated the anti-apoptotic protein Bcl-2 [101,102], leading to elevated mitochondrial membrane permeability, cytochrome c (cyt c) release, and the subsequent activation of caspase-3 and other apoptosis-related proteases, ultimately inducing neuronal apoptosis [103,104]. Furthermore, JNK activation also suppressed the expression and activity of proteins involved in mitochondrial autophagy, such as ULK1 and Beclin-1 [105], resulting in impaired mitochondrial autophagy. Consequently, the damaged mitochondria could not be cleared efficiently, exacerbating mitochondrial dysfunction. In addition, recent studies have found that tau acetylation also contributes to mitochondrial damage. Tau acetylation at K274/K281 not only induced mitochondrial fission by decreasing mitofusion proteins, but also inhibited mitochondrial biogenesis through the reduction of PGC-1a/Nrf1/Tfam levels [106].

### 5.3. Causing Synaptic Dysfunction

Studies have demonstrated that in a mouse model of AD, the number of synapses decreased significantly as tau protein hyperphosphorylation intensified. The structural integrity of both the presynaptic and postsynaptic membranes was compromised, leading to a widened synaptic cleft and an altered distribution of synaptic vesicles. Additionally, the expression levels of proteins associated with synaptic structure, such as synapsin and postsynaptic density protein 95 (PSD-95), were markedly reduced [107,108]. This reduction in protein expression directly impaired synaptic structure and function, thereby decreasing the efficiency of synaptic transmission. Furthermore, the study revealed that in the tau proteinopathy model, key proteins on the synaptic vesicle membrane, including syntaxin and synaptobrevin [109,110], exhibited altered binding affinities for tau proteins. These changes disrupted the synaptic vesicle release mechanism, resulting in a diminished neurotransmitter release and consequently affecting synaptic signaling function. Concurrently, when tau proteins exhibit pathological abnormalities, astrocytes and microglia become activated, releasing inflammatory cytokines such as the tumor necrosis factor-alpha (TNF-α) and interleukin-1 beta (IL-1β). These factors degrade the extracellular matrix surrounding synapses and disrupt the interaction between synapses and glial cells, ultimately contributing to synaptic structural damage and functional impairment [111,112,113].

Abnormalities in tau also impair synaptic plasticity [114]. Synaptic plasticity refers to the capacity of synapses to change their structure and function after stimulation, including long-term potentiation (LTP) and long-term depression (LTD) [115]. In the brains of AD patients, the abnormal aggregation and phosphorylation of tau proteins disrupt signaling pathways associated with synaptic plasticity, such as those mediated by the N-methyl-D-aspartate receptor (NMDAR). NMDAR plays a key role in synaptic plasticity, and abnormalities in its function can lead to impaired LTP and enhanced LTD, thus affecting cognitive functions such as learning and memory [116]. Studies have shown that in both cellular and animal models with tau overexpression, LTP is significantly weakened while LTD is enhanced, suggesting that abnormalities in tau disrupt synaptic plasticity and lead to cognitive dysfunction.

Our previous studies have shown that the signal transducer and activator of transcription (STAT) can regulate tau-induced deficits in synaptic plasticity. Tau accumulation activated JAK2-dependent STAT1 in the animal models of AD, which subsequently leads to STAT1 directly binding to the GAS element of GluN1, GluN2A, and GluN2B promoters, thereby suppressing the expression of NMDARs. STAT3 positively regulates NMDARs transcription by directly binding to the GAS element of GluN1, GluN2A, and GluN2B promoters as STAT1. Additionally, studies have shown STAT3 inactivation in tau-accumulating neurons in AD and frontotemporal dementia [44,117]. The intrinsic enzymatic activity of tau allows it to directly acetylate STAT1, enhancing its binding to STAT3 in the cytoplasm, thus inhibiting the translocation of STAT3 to the nucleus and ultimately reducing STAT3 transcription. Finally, tau activates STAT1 and inhibits STAT3 to reduce the transcription and expression of NMDARs [44,117].

### 5.4. Activating the Neuroinflammatory Response

Neuroinflammatory response plays an important role in the pathogenesis of AD, and an abnormality of tau proteins is one of the key factors that activate a neuroinflammatory response. In the brains of AD patients, abnormal phosphorylation and the aggregation of tau proteins are recognized by microglia and astrocytes, which activate these glial cells and trigger a neuroinflammatory response.

Triggering receptor-expressed on myeloid cells 2 (TREM2), a pivotal risk factor for LOAD [118,119], is associated with tau pathology. TREM2 is a receptor for Aβ and is exclusively expressed by microglia in the brain of mice and humans. In the cerebrospinal fluid (CSF) of AD patients, the R47H (rs75932628) variant of TREM2 or soluble TREM2 has been found to correlate with total or phosphorylated tau (Tr181), respectively [120,121,122].

Microglia are immune cells in the CNS that, under normal conditions, are in a resting state and are primarily responsible for maintaining the homeostasis of the nervous system. When microglia recognize abnormal aggregates of tau proteins, they are activated and transformed into an activated state. Microglia sense pathological tau species via various surface receptors to trigger phagocytosis and/or degradation. Conditional genetic manipulation and genome-wide association studies have identified several immune-related genes (TREM2 [123], CX3CR1, CR1 [124], CD33 [125], ATP-binding cassette transporter (ABCA) 7 [126], and BIN1 [127]) as risk factors for tau pathology and AD, thereby supporting the critical involvement of glial cells and neuroinflammation in tauopathies.

Tau can bind to microglial CX3CR1, initiating the internalization and degradation of tau [128]. Activated microglia release a range of inflammatory factors, such as TNF-α, IL-1β, and interleukin-6 (IL-6), which are cytotoxic and can cause neuronal damage and death. Studies have shown that in the tau transgenic mouse model, microglia are significantly activated, the expression level of inflammatory factors is significantly increased, and the neuronal damage is more severe. In addition, activated microglia also attempt to clear tau protein aggregates through phagocytosis, but in the process, more ROS and other cytotoxic substances may be produced, further aggravating neuronal damage [113].

Astrocytes also play a significant role in neuroinflammatory responses. In the brains of AD patients, abnormalities in tau proteins lead to the activation and proliferation of astrocytes. Activated astrocytes secrete various inflammatory mediators and chemokines, such as nitric oxide (NO) and prostaglandin E2 (PGE2), which further aggravate the neuroinflammatory response [129]. Simultaneously, astrocytes also promote the activation of microglia and the release of inflammatory factors through interactions with microglia. Studies have found that in the brains of AD patients, there exists a complex interaction network between astrocytes and microglia, which activate each other through the release of inflammatory factors and cytokines, forming a vicious circle, leading to the continuous intensification of neuroinflammatory response, ultimately resulting in the loss of a substantial number of neurons and severe cognitive impairment [130].

### 5.5. Interaction with β-Amyloid Protein

Aβ and tau proteins are the two primary pathological markers of AD, and they interact during the onset of AD to jointly promote the progression of the disease [131]. Increasing evidence has demonstrated that a complex interaction exists between Aβ and tau proteins, which plays a key role in the pathogenesis of AD.

Aβ can induce abnormal phosphorylation and the aggregation of tau proteins [132]. The abnormal deposition of Aβ in the brain of AD patients activates various of protein kinases, such as GSK3β [133,134] and Cdk5 [135,136], which catalyze the phosphorylation of tau proteins. Studies have shown that Aβ oligomers can bind to receptors on the surface of neurons and activate intracellular signaling pathways, resulting in increased activity of kinases such as GSK3β and Cdk5, which in turn lead to the abnormal hyperphosphorylation of tau proteins [137]. In addition, Aβ can indirectly promote tau phosphorylation by disrupting intracellular calcium homeostasis [138]. Calcium concentration is an important modulator of S100A9 fibrillation, which can cause a synergistic effect in combination with tau, and is capable of inhibiting the formation of amyloid fibrils [139,140]. In addition, ionic strength has been identified as a crucial factor influencing amyloid aggregation [141]. Abnormally phosphorylated tau proteins are more prone to aggregate to form neurofibrillary tangles, which are neurotoxic and can further damage neurons [142].

Tau proteins also contribute to the production and aggregation of Aβ. Tau proteins can influence the production of Aβ by regulating the metabolic process of amyloid precursor proteins (APPs) [143,144]. Studies have shown that tau proteins can interact with APP to change the processing pathway of APP, making it more likely to produce the neurotoxic Aβ42 subtype [145]. Furthermore, tau proteins promote the aggregation of Aβ, forming oligomers and fibrous structures. The abnormal aggregation of tau proteins disrupts the normal structure and function of neurons, leading to changes in the intracellular environment, which provides favorable conditions for Aβ aggregation [146,147]. In the tau transgenic mouse model, the aggregation of Aβ was significantly increased, indicating that tau proteins can promote the pathological process of Aβ.

The interaction of Aβ and tau proteins also exacerbates neuroinflammatory responses and synaptic dysfunction [146]. The abnormal aggregation of Aβ and tau proteins activates both microglia and astrocytes, triggering a neuroinflammatory response. And the interaction between them further increases the intensity of the inflammatory response. At the same time, Aβ and tau proteins also have a synergistic effect on the damage to synaptic structure and function. Aβ can directly act on synapses and destroy the structure and transmission function of synapses, while the abnormality of tau proteins will lead to changes in the presynaptic and postsynaptic structure, further aggravating synaptic dysfunction [148,149]. Synaptic dysfunction and cognitive decline were more severe in the animal models with both Aβ and tau pathologies, suggesting that the interaction of Aβ and tau proteins exacerbates the pathological course of AD.

## 6. Conclusions and Future Perspectives

This study conducted a comprehensive and in-depth analysis of the PTMs of tau proteins, revealing their key role in neurodegenerative diseases. Various PTMs of tau proteins, including phosphorylation, acetylation, ubiquitination, glycosylation, and methylation, play a crucial regulatory role in the structure, function, and stability of tau proteins under normal physiological conditions, thereby ensuring the proper physiological activities of neurons [150]. In AD, significant abnormalities occur in the PTMs of tau proteins. Tau-mediated mitochondrial damage, synaptic dysfunction, and neuroinflammation lead to neuronal damage and death through multiple mechanisms, thereby triggering the onset and progression of the disease (Figure 3). Once the toxic mechanism of the tau protein is clarified, therapeutic drugs targeting key processes, such as the abnormal modification and aggregation of tau proteins, can be developed [151]. The development of these treatment strategies is expected to bring new therapeutic hope to patients with neurodegenerative diseases, improve the prognosis of patients, enhance the quality of life of patients, and hold significant clinical application value. Studying the toxicity mechanism of tau proteins also facilitates the development of accurate and sensitive biomarkers for the early diagnosis and condition monitoring of neurodegenerative diseases, thereby achieving early intervention and treatment of the disease and further reducing the harm of the disease.

Among the various PTMs of tau, excessive phosphorylation, acetylation, methylation, and glycosylation can lead to the dissociation of tau proteins from microtubules, thereby promoting their aggregation. Conversely, methylation at specific sites exerts an opposing effect by inhibiting tau protein aggregation. For example, K254 methylation may exert neuroprotective effects by maintaining the normal conformation of tau proteins and preventing the abnormal aggregation of tau [57]. In recent years, increasing attention has been paid to the implications of tau protein acetylation for mitochondrial function. The acetylation of a simulated tau protein at the K274 and K281 sites would lead to a decrease in mitochondrial biogenesis, a decrease in the expression of mitochondrial fusion proteins, and an increase in mitochondrial dysfunction. Tau acetylation at K274/K281 induces mitochondrial fission by reducing mitofusion proteins and inhibits mitochondrial biogenesis through the downregulation of PGC-1a/Nrf1/Tfam [106,152]. At present, the mechanisms underlying tau acetylation and mitochondrial damage remain insufficiently explored. This research avenue holds significant potential and may serve as a key focus in future investigations, providing a new therapeutic target for AD.

Recent studies have demonstrated that the phosphorylation and acetylation of tau proteins modulate their interactions with other cellular components. Specifically, the interactions between tau proteins and synaptic or mitochondrial processes are closely associated with neurodegeneration [14,153]. Phosphorylated tau proteins in cerebrospinal fluid serve as a critical diagnostic and prognostic biomarker for AD, correlating with the emergence of clinical symptoms long after the onset of Aβ pathology. Consequently, the early detection and clearance of neurotoxic tau species, either prior to or at the very onset of AD, may represent an effective strategy for preserving neural network integrity and preventing or delaying the progression of AD. Clinical trials on anti-tau vaccines and other immunotherapies that show considerable potential for treating AD in the future are ongoing.

Although significant progress has been made in the research on the toxicity mechanism of tau proteins, numerous issues and challenges still exist. The initiating factors of the abnormal modification of tau proteins have not been fully elucidated. Although it is known that factors such as gene mutations, oxidative stress, and inflammation can promote the abnormal modification and aggregation of tau proteins, the specific mechanisms by which these factors interact and trigger abnormal changes in tau proteins remain unclear. Tau protein modifications exhibit considerable promise as biomarkers and therapeutic targets for neurodegenerative diseases; however, several challenges persist. In terms of biomarkers, it is essential to identify and validate highly specific and sensitive modified biomarkers, develop more precise and user-friendly detection methods, and enhance early diagnostic accuracy. Regarding therapeutic targets, drug discovery focused on enzymes or specific modification sites related to tau protein modifications is still in its early stages. Strengthening the integration of basic research with clinical translation, refining drug design, and improving drug efficacy and safety are critical steps for future progress.

## 7. Reflexivity Statement

Currently, the mechanisms underlying tau acetylation and mitochondrial damage remain incompletely understood. This area of research holds significant potential and could emerge as a focal point for future investigations, thereby providing novel therapeutic targets for AD. Although substantial progress has been made in understanding the toxic mechanisms of tau proteins, numerous challenges and unresolved issues persist. The initiating factors that lead to the abnormal modification of tau proteins have yet to be fully clarified. While it is acknowledged that factors such as genetic mutations, oxidative stress, and inflammation can contribute to the abnormal modification and aggregation of tau proteins, the precise mechanisms through which these factors interact and induce pathological changes in tau proteins remain unclear. All of the aforementioned areas warrant further exploration in future studies.

## Figures and Tables

**Figure 1 biomolecules-15-00824-f001:**
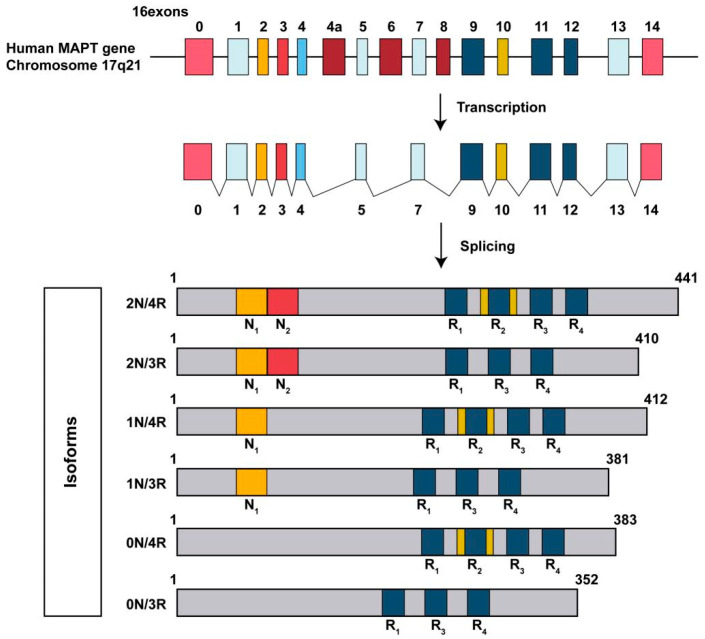
Tau isoforms with alternative splicing. The tau protein consists of 16 exons. Through alternative splicing of these exons, six major isoforms can be generated, each displaying distinct expression patterns in various tissues and developmental stages. The N-terminus serves as the projection domain, comprising the proline-rich region (P-region) and the amino-terminal region enriched in amino acid residues. The C-terminus functions as the microtubule-binding domain, containing the carboxyl-terminal region composed of repetitive polypeptide fragments that bind to tubulin, along with additional amino acid residues. Notably, there are four primary repetitive polypeptide fragments, abbreviated as R1, R2, R3, and R4, respectively.

**Figure 2 biomolecules-15-00824-f002:**
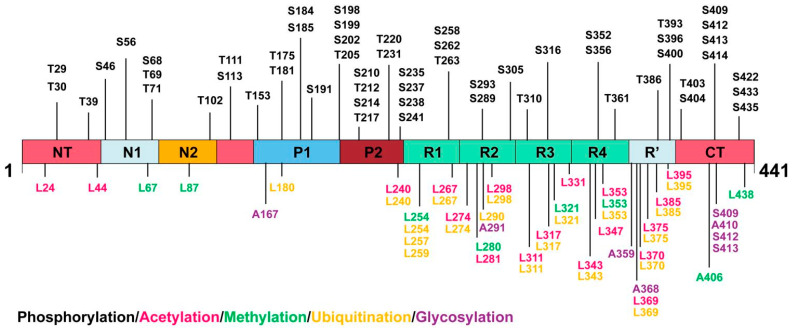
PTMs of the 2N4R tau isoform in AD patients, related to Table 1.

**Figure 3 biomolecules-15-00824-f003:**
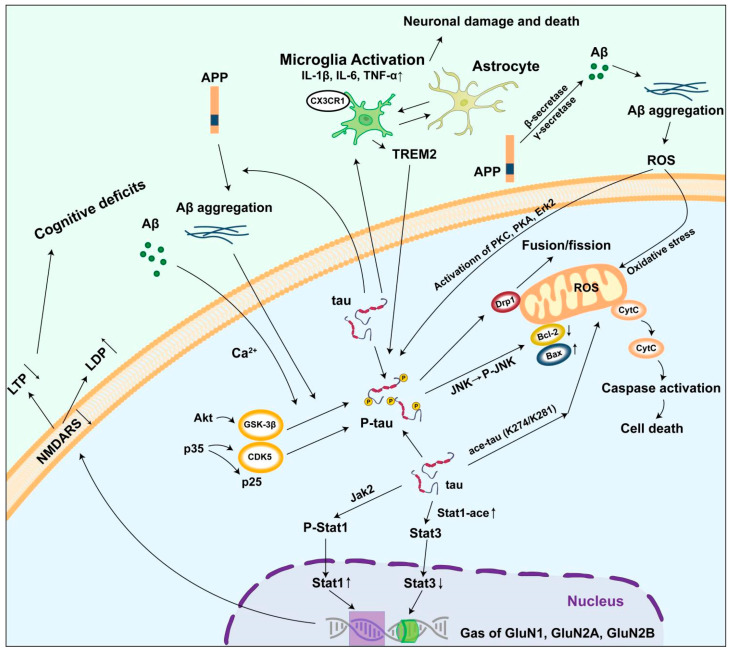
Mechanism of abnormal tau protein triggering AD. Under pathological conditions, APP is cleaved into Aβ by the action of beta- and gamma-secretases. Tau protein undergoes hyperphosphorylation by various kinases. Abnormal PTMs reduce the affinity of tau for microtubule binding, leading to an increase in the levels of free tau protein in the cytoplasm. The abnormal aggregation of both Aβ and tau results in the two major pathological features of AD: extracellular amyloid plaques and intracellular neurofibrillary tangles. The accumulation of free tau promotes the formation of tau aggregates, which subsequently induces mitochondrial dysfunction, impairs synaptic plasticity, and triggers glial-mediated neuroinflammation. Ultimately, these processes culminate in cognitive impairments, including deficits in learning and memory.

**Table 1 biomolecules-15-00824-t001:** PTM sites of the 2N4R tau isoform in AD patients.

Types of PTMs and Corresponding Biochemical Properties	Sites of PTMs			
	N-Terminus	Proline-Rich Domain	MT-Binding Domain	C-Terminus
**phosphorylation:** microtubule stability **hyperphosphorylation:** tau aggregation	Tyr-29, Thr-30, Thr-39, Ser-46, Ser-56, Ser-68, Thr-69, Thr-71, Thr-102, Thr-111, Ser-113	Thr-153, Thr-175, Thr-181, Ser-184, Ser-185, Ser-191, Ser-198, Ser-199, Ser-202, Thr-205, Ser-210, Thr-212, Ser-214, Thr-217, Thr-220, Thr-231, Ser-235, Ser-237, Ser-238, Ser241	Ser-258, Ser-262, Thr-263, Ser-289, Ser-293, Ser-305, Tyr-310, Ser-316, Ser-352, Ser-356,Thr-361	Thr-386, Tyr-393, Ser-396, Ser-400, Thr-403, Ser-404, Ser-409, Ser-412, Ser-413, Ser-414, Ser-422, Ser-433, Thr-435
**acetylation:** causes dissociation from microtubules and prevents tau degradation	Lys-24, Lys-44	Lys-240	Lys-267, Lys-274, Lys-281, Lys-298, Lys-311, Lys-317, Lys-331, Lys-343, Lys-347, Lys-353	Lys-369, Lys-370, Lys-375, Lys-385, Lys-395
**methylation:** promote or prevent tau accumulation	Lys-67, Lys-87		Lys-254, Lys-280, Lys-321, Lys-353	Arg-406, Lys-438
**ubiquitination:** degradation and metabolism of tau		Lys-180, Lys-240	Lys-254, Lys-257, Lys-259, Lys-267, Lys-274, Lys-290, Lys-298, Lys-311, Lys-317, Lys-321, Lys-343, Lys-353	Lys-369, Lys-370, Lys-375, Lys-385, Lys-395
**glycosylation:** reduce solubility, stability, and increase tendency of tau to aggregate		Asn-167	Asn-291, Asn-359, Asn-368	Ser-409, Asn-410, Ser-412, Ser-413

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
