# Peer review of "The Role and Pathogenesis of Tau Protein in Alzheimer’s Disease"

_biomolecules, 2025, doi:10.3390/biom15060824_

Round 1
Reviewer 1 Report
Comments and Suggestions for Authors
The manuscript titled “The role and pathogenesis of tau protein in Alzheimer’s Disease” by Hong, X.; et al. is a Review work where the authors outlined the most recent advances in the field of tau aggregation and impact in Alzheimer’s Disease under the action of postranslational modifications. The Review is coherent and the content properly divided in different subsections. Furthermore, the manuscript is generally well-written and this is a topic of growing interest.
However, it exists some points that need to be addressed (please, see them below detailed point-by-point) to improve the scientific quality of the submitted manuscript paper before this article will be consider for its publication in Biomolecules.
1) Introduction. Could the authors provide quantitative data insights according to the worldwide global burdens of Alzheimer’s disease? This will significantly aid the potential readers to better understand the significance of this devoted research.
2) Then, a schematic representation to highlight the underlying mechanisms and key factors that trigger the tau fibrillization processes and aggregation in amyloid plaques which eventually can lead to neurodegeneration will also benefit the potential readers. This figure could take part in the subsection “3. Basic structure of tau protein” (lines 56-86) and will be complementary with the current Fig. 2 (line 562).
3) “4. PTMs of tau protein” (lines 84-343). In this section it would be advisable to remark those techniques capable to sense the post-translational modification in tau protein family.
4) “5 Mechanism of abnormal tau protein accumulation triggering AD”. “Aβ can indirectly promote tau phosphorylaition by disruption intracellular calcium homeostasis” (lines 498-499). Here, it should be neccesary to mention how this disruption of intracellular calcium homeostasis can induce the fibrillization of other amyloid proteins like the S100A family [1] that can cause a synergistic effect in combination with tau. The media ionic strength is other pivotal factor to consider [2].
[1] https://doi.org/10.3390/biom14091091
[2] https://doi.org/10.3390/ijms222212382
5) “6. Conclusions and Future Perspectives” (lines 523-546). This section perfectly remarks the most relevant outcomes found by the authors in this field and also the promising future prospectives. It may be desirable to add a brief statement to discuss about the potential future action lines to pursue the topic covered in this work.
Author Response
Thank you very much for your comments and suggestions, which have been extremely helpful in improving the manuscript. Below, we provide our responses to each of your comments, and the changes in the manuscript are highlighted in red:
#Comment 1:
Introduction. Could the authors provide quantitative data insights according to the worldwide global burdens of Alzheimer’s disease? This will significantly aid the potential readers to better understand the significance of this devoted research.
#Response 1:
This is an excellent suggestion. We have added the following content in lines 40-45: "Alzheimer's Disease (AD) is a widespread neurodegenerative disorder that represents a substantial threat to the health and quality of life of elderly individuals worldwide. According to the "World Alzheimer's Report 2018" published by Alzheimer's Disease International (ADI), the global prevalence of dementia is estimated to affect at least 50 million individuals. This figure is projected to rise to 152 million by 2050, with approximately 60% to 70% of these cases attributable to AD."
#Comment 2:
Then, a schematic representation to highlight the underlying mechanisms and key factors that trigger the tau fibrillization processes and aggregation in amyloid plaques which eventually can lead to neurodegeneration will also benefit the potential readers. This figure could take part in the subsection “3. Basic structure of tau protein” (lines 56-86) and will be complementary with the current Fig. 2 (line 562).
#Response 2:
We have revised Figure 3, adding key mechanisms and factors that trigger tau protein phosphorylation and Aβ aggregation, based on the original Figure 2. We have also added the following content in lines 645-646: "Under pathological conditions, APP is cleaved into Aβ by the action of beta- and gamma-secretases."
#Comment 3:
"4. Post-translational modifications of tau protein" (lines 84-343). In this section, it is recommended to mention the techniques capable of detecting the post-translational modifications of the tau protein family.
#Response 3:
As suggested, we have added the following content in lines 101-104: "At present, techniques such as western blotting, immunohistochemistry [11], immunofluorescence, proteomics, immunomagnetic exosomal PCR [13], affinity purification-mass spectrometry [14] and liquid chromatography-mass spectrometry [15] have enabled accurate detection of PTMs of tau protein."
#Comment 4:
“5 Mechanism of abnormal tau protein accumulation triggering AD”. “Aβ can indirectly promote tau phosphorylaition by disruption intracellular calcium homeostasis” (lines 498-499). Here, it should be neccesary to mention how this disruption of intracellular calcium homeostasis can induce the fibrillization of other amyloid proteins like the S100A family [1] that can cause a synergistic effect in combination with tau. The media ionic strength is other pivotal factor to consider [2].
[1] https://doi.org/10.3390/biom14091091
[2] https://doi.org/10.3390/ijms222212382
#Response 4:
We have incorporated the following content and references in lines 554-557: "Calcium concentration is an important modulator of the S100A9 fibrillation, which can cause a synergistic effect in combination with tau, and is capable of inhibiting the formation of amyloid fibrils [139,140]. In addition, ionic strength has been identified as a crucial factor influencing its amyloid aggregation [141]."
#Comment 5:
“6. Conclusions and Future Perspectives” (lines 523-546). This section perfectly remarks the most relevant outcomes found by the authors in this field and also the promising future prospectives. It may be desirable to add a brief statement to discuss about the potential future action lines to pursue the topic covered in this work.
#Response 5:
Thank you very much for your meaningful suggestion. We have added the following content in lines 600-626:
Among the various PTMs of tau, excessive phosphorylation, acetylation, methylation, and glycosylation can lead to the dissociation of tau protein from microtubules, thereby promoting its aggregation. Conversely, methylation at specific sites exerts an opposing effect by inhibiting tau protein aggregation.For example,K254 methylation may exert neuroprotective effects by maintaining the normal conformation of tau protein and preventing abnormal aggregation of tau [57]. In recent years, increasing attention has been paid to the implications of tau protein acetylation for mitochondrial function. Acetylation of simulated tau protein at K274 and K281 sites would lead to a decrease in mitochondrial biogenesis, a decrease in the expression of mitochondrial fusion proteins, and an increase in mitochondrial dysfunction. Tau acetylation at K274/K281 induces mitochondrial fission by reducing mitofusion proteins and inhibits mitochondrial biogenesis through the downregulation of PGC-1a/Nrf1/Tfam [106,152]. At present, the mechanisms underlying tau acetylation and mitochondrial damage remain insufficiently explored. This research avenue holds significant potential and may serve as a key focus in future investigations, providing a new therapeutic target for AD.
Recent studies have demonstrated that the phosphorylation and acetylation of tau protein modulate its interactions with other cellular components. Specifically, the interactions between tau protein and synaptic or mitochondrial processes are closely associated to neurodegeneration [14,153]. Phosphorylated tau protein in cerebrospinal fluid serves as a critical diagnostic and prognostic biomarker for AD, correlating with the emergence of clinical symptoms long after the onset of Aβ pathology. Consequently, the early detection and clearance of neurotoxic tau species, either prior to or at the very onset of Alzheimer's disease, may represent an effective strategy for preserving neural network integrity and preventing or delaying the progression of AD. Clinical trials on anti-tau vaccines and other immunotherapies are ongoing that show considerable potential for treating AD in the future.
Reviewer 2 Report
Comments and Suggestions for Authors
This manuscript addresses an increasingly important topic in neurodegeneration: the role of tau post-translational modifications (PTMs) in Alzheimer’s disease (AD). By examining how specific PTMs relate to mitochondrial dysfunction, synaptic impairment, and neuroinflammation, the review contributes to ongoing efforts to clarify molecular mechanisms relevant to AD pathology and therapeutic development. While the manuscript presents a wide-ranging overview of relevant processes involved in the pathology of Alzheimer’s disease, several aspects require further development to improve its structural clarity, methodological transparency, and overall coherence.
- Review Type and Conceptual Framing
The introduction refers to a “comprehensive and systematic analysis” (lines 42–43), yet the manuscript does not follow standard guidelines for systematic reviews (e.g., PRISMA). Instead, it takes the form of a narrative review. For instance, this should be stated explicitly if a narrative review was chosen to allow thematic integration across different mechanisms/pathways.
Additionally, the review would benefit from a more transparent conceptual framework that outlines the connections between tau’s structural domains, specific PTMs, related enzymatic regulators, and the downstream pathological consequences. This would help to organise the content, support interpretation of the literature, and provide a structure for integrating diverse findings. A diagram summarising this framework could help readers understand how the selected studies fit together and where uncertainties remain.
- Search Strategy and Inclusion Criteria
The methodology currently lacks sufficient detail to assess the scope or reproducibility of the literature search. The authors note that PubMed was used as the sole database but do not explain why other databases (e.g., Web of Science, Embase) were excluded. There is also no mention of inclusion/exclusion criteria, screening strategy, or the number of studies reviewed.
To improve methodological transparency, the authors should:
- Justify the choice of database(s).
- Clearly describe the search strategy, including search terms.
- Provide inclusion and exclusion criteria.
- Indicate how studies were selected and prioritised (e.g., recent, high-impact, or foundational studies).
- Report the number of studies screened and included.
This information would help situate the review within current and established academic standards and make it easier for other researchers to evaluate important future research areas.
- Mechanistic Depth and Structural Organisation in PTM Section
Section 4 presents various PTMs of tau and summarises known effects on AD pathology. Table 1 provides a helpful overview of modification sites. However, the section remains largely descriptive and would benefit from clearer organisation and analysis.
To improve this section, the authors should:
- Link PTMs to tau’s structural domains (e.g., N-terminal projection domain, microtubule-binding repeats);
- Describe how specific modifications influence tau’s biochemical properties (e.g., solubility, aggregation, degradation);
- Highlight whether modifications act independently or in synergy;
- Provide individual figures mapping PTM sites to tau structure, alongside downstream effects such as aggregation or organelle disruption for each PTM described, reducing the wording of each subsection.
- Additionally, a final integrative figure summarising the section overall would be valuable to highlight how these PTMs interact, differ in pathogenic impact, and may converge on shared disease mechanisms or therapeutic targets.
These additions would strengthen the interpretation of mechanistic data and provide a more straightforward path from molecular changes to pathological outcomes.
- Integration of PTMs into the Tau–β Amyloid Interaction
The section describes how the interaction contributes to disease progression, but does not sufficiently integrate the role of PTMs in this interaction. For example, it remains unclear how PTMs may influence tau’s interaction with β Amyloid protein and disease progression.
Integrating previously discussed PTMs into this section would improve consistency across the manuscript and highlight potential mechanisms of interventions and the development of therapeutic strategies.
Finally, the section Conclusions and future perspectives reiterates that abnormal PTMS play a central role in tau pathology. A better understanding of these PTMS and the enzymes involved may provide treatment opportunities. Still, it lacks a unifying statement tying tau’s structural transformation, PTMS, and downstream pathology into a cohesive narrative and how this review advances the field’s understanding of tau’s role in Alzheimer’s disease pathogenesis and what future research directions are most urgent.
- Conclusion and Future Directions
The conclusion reiterates the relevance of abnormal PTMs in tau pathology but lacks a cohesive summary of the key findings and a clear direction for future research. The authors should connect the review’s main points, tau structure, PTMs, and downstream effects, into a unifying perspective that helps guide next research steps in the field.
Consider including:
- A summary of which PTMs are most consistently linked to pathogenic changes;
- Discussion of current knowledge gaps (e.g., functional effects of less-studied PTMs);
- Suggestions for how future studies might better characterise the timing, combinations, or reversibility of PTMs in disease progression.
While Figure 2 contributes a basic schematic aligned with the paper’s general conclusions, it falls short as an integrative or mechanistically nuanced synthesis tool.
Comments on the Quality of English LanguageThe scientific clarity is adequate. Some professional English editing might improve, considering the quality and readability of the paper, something that is expected, considering that the authors are from China and should be addressed during editorial review.
-
Fluency and style: Needs revision for smoother, more professional English
Author Response
Thank you for your valuable comments. Your suggestions have been extremely helpful in improving the manuscript. All modifications are highlighted in red throughout the paper. Below is our response to your comments:
#Comment 1:
Review Type and Conceptual Framing ...
#Response 1:
Thank you for your insightful suggestion.
(1) This review is a narrative review, focused primarily on the role of tau protein in Alzheimer’s Disease and its mechanisms. The content is based on the knowledge and experience our team has accumulated in this area.
(2) We have added new content to Figure 2 and improved Figure 3 to clarify the interrelationships between tau protein domains, specific post-translational modifications, and regulatory enzymes in the pathology of AD, which leads to the two major pathological features of AD. In lines 648-650, we added: “The abnormal aggregation of Aβ and tau collectively leads to the two major pathological features of AD: extracellular amyloid plaques and intracellular neurofibrillary tangles.”
#Comment 2:
Search Strategy and Inclusion Criteria ...
#Response 2:
Thank you for your comment! As mentioned in Response 1, this is a narrative review. To avoid confusion, we have removed the content related to "2. Search strategy" from the manuscript.
#Comment 3:
Mechanistic Depth and Structural Organisation in PTM Section ...
#Response 3:
Following your suggestion, we have added “Types of PTMs and corresponding biochemical properties” content to Table 1 and mapped the PTMs sites to the tau structure in Figure 2. We have also further improved Figure 3, which now summarizes the content of this section, highlighting how different PTMs interact, their pathogenic differences, and how they converge on shared disease mechanisms or therapeutic targets.
#Comment 4:
Integration of PTMs into the Tau–β Amyloid Interaction ...
#Response 4:
This is an excellent suggestion! We have made additions in the following sections:
- Lines 554-557: "Calcium concentration is an important modulator of the S100A9 fibrillation, which can cause a synergistic effect in combination with tau, and is capable of inhibiting the formation of amyloid fibrils [139,140]. In addition, ionic strength has been identified as a crucial factor influencing its amyloid aggregation [141]."
- Lines 166-175: "The phosphorylation sites of tau protein may play opposing roles in physiological and pathological processes. For instance, phosphorylation at the Ser202 and Thr231 sites of tau promotes its aggregation, neurofibrillary tangle formation, and synaptic dysfunction [34]. Additionally, a study demonstrated that a humanized monoclonal antibody targeting the phosphorylation of Thr231 in tau could reduce neurofibrillary tangle formation and tau seeding activity while enhancing synaptic and cognitive functions [35]. These findings suggest that distinct tau phosphorylation sites play differential roles in tau-related pathology and cognitive impairment. Consequently, selecting the appropriate antigenic epitope when developing therapies targeting tau phosphorylation may enhance both the safety and efficacy of such interventions."
- Lines 226-235: "TauKQhigh mice expressing human tau with lysine-to-glutamine mutations that mimic acetylation at K274 and K281 have been previously shown to exhibit memory impairments and deficits in long-term potentiation[36]. The interesting observation that the hypothalamus of tauKQhigh mice exhibits significantly increased axonal neurodegeneration, in contrast to the previously observed neuroprotective effects in this region following traumatic brain injury [49]. The monoclonal antibody Y01 specifically targeting tau protein Lys280 has demonstrated significant therapeutic effects on tau protein accumulation, reproduction and cognitive deficits in tau transgenic mouse models [50]. Therefore, targeting tau acetylation represents a promising novel therapeutic strategy for treating tauopathies in humans."
#Comment 5:
Conclusion and Future Directions ...
#Response 5:
Thank you very much for your suggestion. We have added content in lines 600-626:
Among the various PTMs of tau, excessive phosphorylation, acetylation, methylation, and glycosylation can lead to the dissociation of tau protein from microtubules, thereby promoting its aggregation. Conversely, methylation at specific sites exerts an opposing effect by inhibiting tau protein aggregation.For example,K254 methylation may exert neuroprotective effects by maintaining the normal conformation of tau protein and preventing abnormal aggregation of tau [57]. In recent years, increasing attention has been paid to the implications of tau protein acetylation for mitochondrial function. Acetylation of simulated tau protein at K274 and K281 sites would lead to a decrease in mitochondrial biogenesis, a decrease in the expression of mitochondrial fusion proteins, and an increase in mitochondrial dysfunction. Tau acetylation at K274/K281 induces mitochondrial fission by reducing mitofusion proteins and inhibits mitochondrial biogenesis through the downregulation of PGC-1a/Nrf1/Tfam [106,152]. At present, the mechanisms underlying tau acetylation and mitochondrial damage remain insufficiently explored. This research avenue holds significant potential and may serve as a key focus in future investigations, providing a new therapeutic target for AD.
Recent studies have demonstrated that the phosphorylation and acetylation of tau protein modulate its interactions with other cellular components. Specifically, the interactions between tau protein and synaptic or mitochondrial processes are closely associated to neurodegeneration [14,153]. Phosphorylated tau protein in cerebrospinal fluid serves as a critical diagnostic and prognostic biomarker for AD, correlating with the emergence of clinical symptoms long after the onset of Aβ pathology. Consequently, the early detection and clearance of neurotoxic tau species, either prior to or at the very onset of Alzheimer's disease, may represent an effective strategy for preserving neural network integrity and preventing or delaying the progression of AD. Clinical trials on anti-tau vaccines and other immunotherapies are ongoing that show considerable potential for treating AD in the future.
#Comment 6:
Comments on the Quality of English Language ...
#Response 6:
We have edited the entire manuscript for English language improvement to enhance the quality and professionalism of the paper.
Round 2
Reviewer 2 Report
Comments and Suggestions for Authors
The authors have addressed many previous comments, and the manuscript has improved in several areas. However, key elements still require revision to align with a narrative review's stated aims and scholarly standards.
While the response letter clarifies that the work is intended as a narrative review grounded in the authors’ expertise, the manuscript (Page 2, line 53) refers to it as a “comprehensive and systematic analysis.” This language is inconsistent with the narrative review format and should be revised for accuracy.
Moreover, the manuscript does not explain the rationale for adopting a narrative approach. This is essential for positioning the review, especially given the likely complexity, interdisciplinarity, or conceptual diversity of the literature. A brief justification should be integrated into the introduction to frame the methodology appropriately.
The original methodology section has been removed rather than adapted. Even in narrative reviews, methodological transparency remains important. The authors should briefly describe how literature was identified, which sources were consulted, and how studies were selected or prioritised (e.g., relevance, scope, conceptual contribution).
Finally, a brief reflexivity statement would enhance the review’s scholarly integrity. Narrative synthesis benefits from acknowledging how the authors’ disciplinary backgrounds or prior knowledge may have shaped the interpretation of findings. Similarly, commenting on how thematic sufficiency or saturation was assessed would strengthen the rigour of the review process.
Author Response
Thank you very much for your comments and suggestions. We will provide a one-on-one response and mark the modified parts in red in the manuscript.
#Comment 1:
While the response letter clarifies that the work is intended as a narrative review grounded in the authors’ expertise, the manuscript (Page 2, line 53) refers to it as a “comprehensive and systematic analysis.” This language is inconsistent with the narrative review format and should be revised for accuracy.
#Response 1:
To ensure the accuracy and consistency of this review, the phrase “comprehensively and systematically” was removed from Page 2, line 53 of the manuscript.
#Comment 2:
Moreover, the manuscript does not explain the rationale for adopting a narrative approach. This is essential for positioning the review, especially given the likely complexity, interdisciplinarity, or conceptual diversity of the literature. A brief justification should be integrated into the introduction to frame the methodology appropriately.
#Response 2:
Thank you for your comment! In response to your suggestion, we have added the following content in lines 55-57: "This review employs a narrative approach to synthesize our team's accumulated knowledge regarding the relationship between tau protein and AD."
#Comment 3:
The original methodology section has been removed rather than adapted. Even in narrative reviews, methodological transparency remains important. The authors should briefly describe how literature was identified, which sources were consulted, and how studies were selected or prioritised (e.g., relevance, scope, conceptual contribution).
#Response 3:
(1) To clarify the writing logic and search strategy for this review, we have reintroduced a brief methodology section in the manuscript, as per your suggestion.
(2) Since the review focuses on "The role and pathogenesis of tau protein in Alzheimer's Disease," the PubMed medical database is sufficient for the literature search scope on this topic. The articles were prioritized based on their relevance to the keywords, as well as their authority, high impact, and timeliness. We have added this information in lines70-72: "The studies were selected and cited prioritizing their relevance to the topic of this review. Subsequently, their authority, high impact factor, and timeliness were comprehensively evaluated."
#Comment 4:
Finally, a brief reflexivity statement would enhance the review’s scholarly integrity. Narrative synthesis benefits from acknowledging how the authors’ disciplinary backgrounds or prior knowledge may have shaped the interpretation of findings. Similarly, commenting on how thematic sufficiency or saturation was assessed would strengthen the rigour of the review process.
#Response 4:
This is an excellent suggestion. We have added the following reflexivity statement in lines 660-671: "7. Reflexivity statement
Currently, the mechanisms underlying tau acetylation and mitochondrial damage remain incompletely understood. This area of research holds significant potential and could emerge as a focal point for future investigations, thereby providing novel therapeutic targets for AD. Although substantial progress has been made in understanding the toxic mechanisms of tau protein, numerous challenges and unresolved issues persist. The initiating factors that lead to the abnormal modification of tau protein have yet to be fully clarified. While it is acknowledged that factors such as genetic mutations, oxidative stress, and inflammation can contribute to the abnormal modification and aggregation of tau protein, the precise mechanisms through which these factors interact and induce pathological changes in tau protein remain unclear. All of the aforementioned areas warrant further exploration in future studies."
